# MB2C: Multimodal Bidirectional Cycle Consistency for Learning Robust Visual Neural Representations

Yayun Wei*
College of Information Engineering
Shanghai Maritime University
Shanghai, China
202230310109@stu.shmtu.edu.cn

Lei Cao†
College of Information Engineering
Shanghai Maritime University
Shanghai, China
lcao@shmtu.edu.cn

Hao Li*
College of Information Engineering
Shanghai Maritime University
Shanghai, China
202230310122@stu.shmtu.edu.cn

Yilin Dong
College of Information Engineering
Shanghai Maritime University
Shanghai, China
ylDong@stu.shmtu.edu.cn

## Abstract

Decoding human visual representations from brain activity data is a challenging but arguably essential task with an understanding of the real world and the human visual system. However, decoding semantically similar visual representations from brain recordings is difficult, especially for electroencephalography (EEG), which has excellent temporal resolution but suffers from spatial precision. Prevailing methods mainly focus on matching brain activity data with corresponding stimuli-responses using contrastive learning. They rely on massive and high-quality paired data and omit semantically aligned modalities distributed in distinct regions of the latent space. This paper proposes a novel Multimodal Bidirectional Cycle Consistency (MB2C) framework for learning robust visual neural representations. Specifically, we utilize dual-GAN to generate modality-related features and inversely translate back to the corresponding semantic latent space to close the modality gap and guarantee that embeddings from different modalities with similar semantics are in the same region of representation space. We perform zero-shot tasks on the ThingsEEG dataset. Additionally, we conduct EEG classification and image reconstruction on both the ThingsEEG and EEGCVPR40 datasets, achieving state-of-the-art performance compared to other baselines.

## CCS Concepts

• **Human-centered computing → Human computer interaction (HCI)**; **HCI theory, concepts and models**;

---

*Both authors contributed equally to this research.
†Corresponding author.

---

## Keywords

Neural decoding, Multimodal learning, Cycle consistency, Zero-shot learning, Image Reconstruction

**ACM Reference Format:**
Yayun Wei, Lei Cao, Hao Li, and Yilin Dong. 2024. MB2C: Multimodal Bidirectional Cycle Consistency for Learning Robust Visual Neural Representations. In *Proceedings of the 32nd ACM International Conference on Multimedia (MM '24), October 28-November 1, 2024, Melbourne, VIC, Australia.* ACM, New York, NY, USA, 9 pages. https://doi.org/10.1145/3664647.3681292

## 1 Introduction

Understanding the complex mechanism of how the human brain perceives the world is fundamental to cognitive science and the development of deep learning. We argue that the behavior of humans recognizing objects and learning concepts are fundamentally linked to brain activity. This connection is inherently multimodal [1], indeed, there exists lots of evidence from neuroscience suggesting that cognitive representations are cross-modal. For instance, the same evoke appears in the same brain neurons when humans watch a picture and hear someone's name or voice [2]. The correlation between brain activity signals (such as electroencephalography (EEG) signals) and external stimuli (such as visual and aural stimuli) elucidates how the brain perceives, processes, organizes, and understands the information from the external environment.

Decoding brain activity signals aims to deepen our understanding of the brain and create user-friendly brain-computer interfaces (BCIs). EEG has garnered significant attention from researchers due to its non-invasive nature, high temporal resolution, and portability. Notably, learning robust visual neural representations from EEG signals has been a focal point of research [3, 4]. Recently, to explore the relationships between different modalities for decoding brain activity, Du et al. [5] constructed three trimodal datasets and pioneered to present a generic neural decoding method named BraVL, which employs intra- and inter-modality mutual information regularization principles to achieve multimodal learning of brain-visual-linguistic features. Subsequently, Song et al. [6] proposed NICE, a self-supervised framework for learning visual representations from EEG signals, particularly for object recognition. Specifically, they aligned these two modalities by leveraging contrastive learning

to compute their similarity. Singh et al. [7] utilized a two-stage method that extracts robust EEG features and subsequently used learned representations for image generation and classification.

However, these approaches above neglect the heterogeneity of modalities, as embeddings of different modalities are distributed in completely independent subspaces, increasing the difficulty of modal alignment [8]. To address this issue, we require a robust mechanism to close the modality gap and enable embeddings from different modalities with similar semantics to distribute in the same region of the representation space. Inspired by this motivation, we introduce Multimodal Bidirectional Cycle Consistency (MB2C), which enforces the generated features to approximate the distribution of realistic samples by learning and utilizing a cycle consistency loss between synthetic EEG/image representations and ground truth data, narrowing the gap between modalities. Furthermore, we combine MB2C with contrastive learning to jointly constrain the model's training, achieving cross-modal alignment between EEG and images.

Beyond the challenges mentioned, due to privacy concerns and the expensive nature of collection equipment, obtaining large datasets of recorded brain activity signals poses challenges, resulting in a scarcity of paired (stimulus-response) training data. When there is a lack of sufficient paired data, models often suffer from poor generalization, especially in zero-shot tasks [9]. To address this, we introduce mixup-based data augmentation methods [10, 11] on zero-shot learning (ZSL) to compensate for the scarcity of paired brain-visual training data.

**Contributions.** In summary, our main contributions are listed as follows:

- We propose a novel multimodal framework for learning robust visual neural representations from EEG-based brain activity.
- The Multimodal Bidirectional Cycle Consistency (MB2C) proposed is also utilized to narrow the gap between modalities and ensure that embeddings from different modalities can be distributed in the same region of the representation space.
- We combine MB2C with contrastive learning to jointly constrain the model's training, effectively achieving cross-modal alignment between EEG and images.
- We perform zero-shot recognition on the ThingsEEG dataset and EEG classification and image reconstruction tasks on the EEGCVPR40 dataset, achieving state-of-the-art performance compared to other baselines.

## 2 Related Works

### 2.1 Multimodal Learning

Our research focuses on multimodal learning. Human perception is inherently multimodal, constituting a complex process involving close collaboration between the brain and multiple sensory systems. Some studies try to decode brain activity and translate it into understandable outputs such as natural language or speech. Dewave et al. [12] translated brain dynamics into natural language using a quantized variational encoder to derive codex encoding and aligned it with pre-trained language models. Guo et al. [13] introduced Dual-Dual GAN that translates neural activity to speech. Other studies focused on sensory stimuli comprising visual stimuli (such as image and video). Kupershmidt et al. [14] proposed a self-supervised

method by utilizing cycle consistency over Encoding-Decoding natural videos to achieve natural-movie reconstruction from brain activity. Xia et al. [15] and Takagi et al. [16], both based on diffusion model to reconstruct images via functional magnetic resonance imaging (fMRI).

### 2.2 Zero-shot Learning (ZSL)

Zero-shot Learning is a task that recognizes unseen classes using a model trained on seen classes. The critical point is to learn how to extract useful features from different modalities and map them to a representation space that reflects class semantics. A taxonomy of ZSL methods can be categorized into two types: embedding-based methods and generative methods [17]. Jiang et al. [18] proposed a method Transferable Contrastive Network (TCN) that contrasts images from different classes to exploit the consistency of their class similarities. Liu et al. [19] introduced a novel goal-oriented gaze estimation module to predict actual human gaze location that gets discriminative visual areas for recognizing novel classes. In the context of ZSL using generative methods, Gao et al. [20] proposed f-VAEGAN, which combined VAE and GAN for discriminative feature synthesis, and Narayan et al. [21] designed a feedback module in a VAE-GAN architecture. The modalities (such as text and image) used in previous methods are more easily collected than brain activity data. In our work, both brain activity and corresponding visual stimuli are provided for training, while during the testing phase, only brain activity data of novel classes are available to perform the unseen class neural decoding task.

### 2.3 Generating Images from Brain Activity Data

Generative Adversarial Network (GAN) [22] was proposed to train a generative model from atributive data distribution. The conventional methods used to generate unimodal data (such as text, image, audio, EEG), EEG-GAN [23], a modification of Wasserstein GANs, was introduced to generate naturalistic EEG data. Recently, Singh et al. [24] proposed a contrastive learning-based method to extract features for conditional GAN (cGAN) to generate images from brain activity signals on EEGCVPR40 dataset that was used in the initial work [25]. Following this, Singh et al. [7] continued introducing a two-stage framework, EEGStyleGAN-ADA, using the CLIP [26] for joint representation learning and transforming the unseen images into EEG feature space with a pre-trained image encoder to reconstruct the images. Unlike GAN-based models, Bai et al. [27] introduced the Diffusion model DreamDiffusion based on the concept of Mind-vis [28], achieving excellent performance. Additionally, Benchetrit et al. [29] proposed a MEG-based model called MEG-BD to ensure the temporal and spatial resolution of the generated images.

## 3 Methodology

### 3.1 Overview

The dataset is defined as $D = \{(x_e, x_v, y) | x_e \in X_e, x_v \in X_v, y \in Y\}$, where $X_e$ denotes the EEG data, $X_v$ represents the corresponding image data and $Y$ denotes the set of class labels. We input stimulus-response pairs, which consist of images and EEG signals, into the model. The brain and image encoder extract features from their respective modalities, resulting in features $f_e(x_e)$ and $f_v(x_v)$.

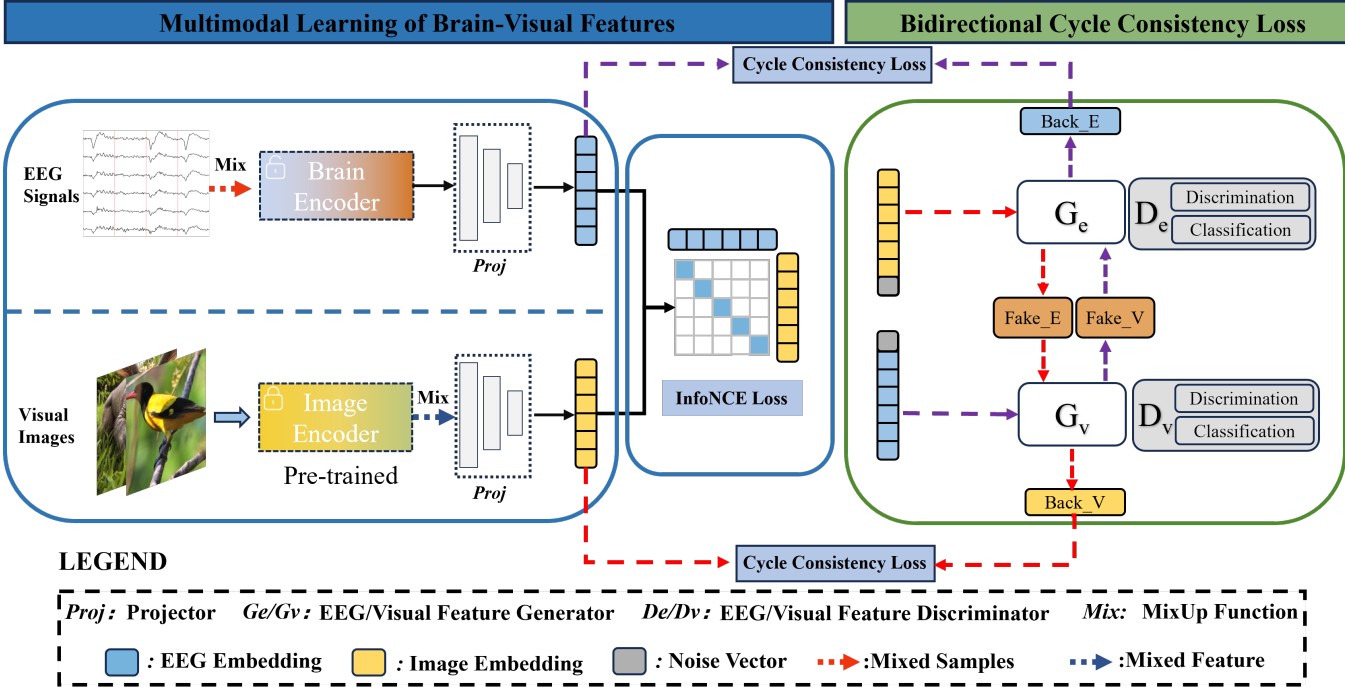

**Figure 1: The overview of our proposed framework MB2C. MB2C mainly contains two parts: Multimodal Learning of Brain-Visual Features and Bidirectional Cycle Consistency Loss. Given paired EEG and image inputs are processed using mixup-based augmentation, then fed into the brain and image encoders to acquire EEG and visual embeddings respectively. Both embeddings are input into a Bidirectional Classified WGAN to generate inverted features. This process utilizes cycle consistency loss for mapping back to the original embeddings, assisted by InfoNCE loss.**

In ZSL [30], the dataset $D$ is partitioned into distinct seen and unseen subsets, serving as the training set $D^s$ and testing set $D^u$, respectively. Similarly, the label set $Y$ is divided into mutually exclusive seen $Y^s$ and unseen $Y^u$ subsets. It is crucial to emphasize that the unseen class labels $Y^u$ are unavailable in the training phase; they are used exclusively during testing. ZSL aims to predict images corresponding to previously unseen EEG data by leveraging information from EEG introduced during the testing phase.

As illustrated in Figure 1, we present multimodal bidirectional cycle consistency (MB2C). Firstly, we introduce mixup-based data augmentation methods and multimodal learning of Brain-Visual features. Subsequently, we introduce bidirectional cycle consistency loss based on bidirectional classified WGAN (BCWGAN).

## 3.2 Mixup-based Data Augmentation

In ZSL, due to the training set only containing EEG-image pairs for seen classes, using a mapping learned from seen data without accommodating the unseen data can lead to a model bias towards these seen classes. To address this issue, we employ a mixup-based data augmentation method [10], which generates diverse samples by performing convex combinations of two instances and their respective labels, preventing deep models from overfitting. Since image features are already preserved using a pre-trained image encoder, and recent studies suggest the superior performance of mixing features over original images [11], we opt for a feature-level mixup approach, where we randomly combine different class image features from the current batch to synthesize new image data. Additionally, for EEG data with a low signal-to-noise ratio, we apply data-level mixup to blend EEG raw data of different classes. Given randomly two different class EEG-image sample pairs $(x_e^i, x_v^i, y^i)$ and $(x_e^j, x_v^j, y^j)$ from our training data, the augmentation process using the following formulation:

$$x_e^k = \gamma \, x_e^i + (1 - \gamma) x_e^j, \tag{1}$$

$$x_v^k = \gamma \, f_v(x_v^i) + (1 - \gamma) f_v(x_v^j), \tag{2}$$

$$y^k = \gamma \, y^i + (1 - \gamma) y^j, \tag{3}$$

The mixing ratio $\gamma \in [0, 1]$ is randomly sampled from a Beta distribution. Values of $\gamma$ close to 0 or 1 result in generated samples more similar to one of the input data, while values close to 0.5 lead to a more balanced mixture of the data. The generated image features and EEG data are fed into the visual projector and brain encoder, respectively, participating in the ensuing training process.

## 3.3 Multimodal Learning of Brain-Visual Features

Generating high-quality and discriminative images from EEG features is challenging due to the high noise in EEG embedding compared to the attribute vector (semantic embedding) for class and

the inherent problems of mode collapse and convergence failure in GANs. Inspired by [31], we employ a cross-modal contrastive learning approach to align images and EEG, maximizing the similarity between EEG signals and corresponding visual stimuli. Specifically, we input image and EEG signals stimuli-response pairs into the model, employing an image encoder and a brain encoder to learn image and EEG embeddings, respectively. Subsequently, we utilize the InfoNCE loss [32] as the objective function to maximize the cosine similarity between matching pairs while minimizing the cosine similarity between non-matching pairs. This cross-modal alignment approach at the instance level, rather than the pixel level, ensures distinguishable model outputs, enabling accurate retrieval of visual stimuli corresponding to EEG encodings. In a batch containing $N$ pairs of EEG-image, denote the embedding of the $i$th image as $z_i^I$ and that of the $i$th EEG as $z_i^E$ in the same bath. The InfoNCE loss for images is formulated as:

$$L_{I2E} = -\frac{1}{N}\sum_{i=0}^{N}\log\frac{\exp(z_i^I \cdot z_i^E/\tau)}{\sum_{k=0}^{N}\exp(z_i^I \cdot z_k^E/\tau)}, \qquad (4)$$

where similarity is measured by dot product and the temperature hyper-parameter $\tau$ is used for adjusting distribution probability. Symmetrically, we define the loss for the EEG as follows:

$$L_{E2I} = -\frac{1}{N}\sum_{i=0}^{N}\log\frac{\exp(z_i^E \cdot z_i^I/\tau)}{\sum_{k=0}^{N}\exp(z_i^E \cdot z_k^I/\tau)}, \qquad (5)$$

The total loss of the InfoNCE thus expresses:

$$L_{InfoNCE} = \frac{1}{2}(L_{I2E} + L_{E2I}), \qquad (6)$$

**Brain and image encoder.** Any parameter model based on deep neural networks (DNNs) can encode raw EEG signals into EEG embedding. Currently, researchers propose several excellent models for analyzing EEG data, and in this study, we use the TSConv [6] as the brain encoder for extracting EEG features. Additionally, we introduce a brain projector to convert the EEG embedding into image embedding of the same dimension. Similarly, a robust image encoder can extract visually distinctive features from images. We use pre-trained CLIP [26] as the image encoder to extract visual features and save them to speed up the training of our model. Similarly, we feed the extracted image features into a visual projector during training.

### 3.4 Bidirectional Cycle Consistency Loss

*3.4.1 Bidirectional Classified Wasserstein GAN (BCWGAN).* Building on the inspiration of f-CLSWGAN [33], we propose a bidirectional classified Wasserstein GAN (BCWGAN). Differing from f-CLSWGAN, BCWGAN consists of two GAN modules: an EEG feature generation network $E = \{G_e, D_e\}$ and a visual feature generation network $V = \{G_v, D_v\}$. Specifically, both generative networks consist of a generator $G$ and a discriminator $D$ participating in an adversarial min-max game, generating synthetic features from a predefined distribution.

**Visual feature generation network.** The visual generator $G_v$ uses EEG embedding $f_e(x_e)$ obtained from the brain encoder and a random noise distribution $z_1 \in \mathbb{R}^{d_z}$ sampled from a Gaussian distribution $N(0, 1)$ to generate fake visual features intended to deceive the visual feature discriminator $D_v$. To distinguish between

the synthetic features and the real features, the visual discriminator $D_v$ employs a fully connected (FC) layer for binary classification. Additionally, introducing classification loss in the discriminator has demonstrated significant potential in auxiliary classifier GANs [34, 35]. Therefore, we incorporate an auxiliary classifier into $D_v$ for classifying input samples into their respective image categories. Due to challenges associated with traditional GANs [22] such as training difficulties and the inability of generator and discriminator losses to guide training progress, we opt for a more stable training method known as WGAN [36]. Its loss function is defined as follows:

$$\begin{aligned}L_{WGAN1} = \min_{\theta_{G_v}}\max_{\theta_{D_v}} &\ \mathbb{E}[D_v(f_v(x_v), \theta_{D_v})] \\ &- \mathbb{E}[D_v(G_v(f_e(x_e), z_1; \theta_{G_v}); \theta_{D_v}] \\ &- \beta\mathbb{E}[(\left\|\nabla_{f_v(\hat{x}_v)}D_v(f_v(\hat{x}_v))\right\|_2 - 1)^2], \qquad (7)\end{aligned}$$

$$\begin{aligned}L_{BCWGAN1} = \min_{\theta_{G_v}}\max_{\theta_{D_v}} &\ L_{WGAN1} - \alpha(L_{CLS1}(D_v(f_v(x_v), \theta_{D_v})) \\ &+ L_{CLS1}(D_v(G_v(f_e(x_e), z_1; \theta_{G_v})))). \qquad (8)\end{aligned}$$

In the formula, $\mathbb{E}[.]$ represents the mathematical expectation, $\theta_{G_v}$ and $\theta_{D_v}$ respectively denote the parameters of the visual generator and visual discriminator. In the last term of equation (7), a gradient penalty term is employed to impose the Lipschitz constraint [37]. Specifically, $f_v(\hat{x}_v) = \gamma f_v(x_v) + (1 - \gamma)G_v(f_e(x_e), z_1; \theta_{G_v})$, where $\gamma \sim U(0, 1)$. The variables $\beta$ and $\alpha$ are two hyperparameters, with $\beta$ representing the gradient penalty factor, and $\alpha$ is utilized to weigh the contribution to the visual feature classification loss $L_{CLS1}$. Here, $L_{CLS1}$ denotes the cross entropy loss between the visual features and their corresponding real labels $y$.

**EEG feature generation network.** Considering the introduced multimodal cycle consistency loss, which involves generating EEG features from visual features, we propose the EEG Feature Generation Network $E = \{G_e, D_e\}$, analogous to $V = \{G_v, D_v\}$, $E$ consists of an EEG feature generator $G_e$ and an EEG feature discriminator $D_e$. First, we concatenate the image embedding with a randomly sampled noise vector $z_2$ from a standard Gaussian distribution. The generator $G_e$ takes the concatenated vector of EEG features and noise, represented as $[x_v, z_2]$, as the input to generate synthesized EEG features in the latent space. Meanwhile, the discriminator $D_e$ discriminates between the real EEG features and synthetic features from $G_e(f_v(x_v), z_2; \theta_{G_e})$. To improve the accuracy of the EEG feature generation, we introduce the EEG feature classification loss $L_{CLS2}$, which guides the EEG generation toward the specified direction. Therefore, the objective function can be formulated as follows:

$$\begin{aligned}L_{WGAN2} = \min_{\theta_{G_e}}\max_{\theta_{D_e}} &\ \mathbb{E}[D_e(f_e(x_e), \theta_{D_e})] \\ &- \mathbb{E}[D_e(G_e(f_v(x_v), z_2; \theta_{G_e}); \theta_{D_e}] \\ &- \beta\mathbb{E}[(\left\|\nabla_{f_e(\hat{x}_e)}D_e(f_e(\hat{x}_e))\right\|_2 - 1)^2], \qquad (9)\end{aligned}$$

$$\begin{aligned}L_{BCWGAN2} = \min_{\theta_{G_e}}\max_{\theta_{D_e}} &\ L_{WGAN2} \\ &- \alpha(L_{CLS2}(D_e(f_e(x_e), \theta_{D_e})) \\ &+ L_{CLS2}(D_e(G_e(f_v(x_v), z_2; \theta_{G_e}))). \qquad (10)\end{aligned}$$

where equation (9) denotes Wasserstein adversarial loss, the first two terms calculate the Wasserstein distance between the synthesized EEG feature distribution and the real EEG feature distribution. The last term computes the gradient penalty by linearly interpolating between the real and generated EEG features. Equation (10) introduces the EEG feature supervised classification loss to ensure that EEG features are as discriminative as visual features.

*3.4.2 Multimodal Bidirectional Cycle Consistency Loss.* In recent years, numerous studies have successfully employed GAN to transform extracted EEG features into corresponding image features [7, 24, 38]. However, existing methods solely rely on unidirectional generative networks, specifically transforming EEG into images. This limitation of the network architecture does not guarantee the accurate representation of the generated visual features corresponding to the underlying brain activities. To address this issue, we propose a bidirectional multimodal cycle consistency loss designed to encourage the visual features generated from EEG features to transform back to their respective EEG embeddings. Similarly, the EEG features generated from image features should also be capable of transforming back to their corresponding image embeddings. After $G_v$ and $G_e$ have generated the corresponding reconstructed features, we compute the bidirectional cycle consistency loss to update the parameters of $G_v$ and $G_e$. In this way, $G_v$ and $G_e$ can perform feature reconstruction while retaining information from the original data. Formally, the loss is defined as follows:

$$L_{cyc1}(\theta_{G_v}, \theta_{G_e}) = \frac{1}{N} \sum_{i=1}^{N} ||G_e(G_v(f_e(x_e), z_1, \theta_{G_v}), z_1; \theta_{G_e}) - f_e(x_e)||^2,$$

(11)

$$L_{cyc2}(\theta_{G_v}, \theta_{G_e}) = \frac{1}{N} \sum_{i=1}^{N} ||G_v(G_e(f_v(x_v), z_2, \theta_{G_e}), z_2; \theta_{G_v}) - f_v(x_v)||^2,$$

(12)

$$L_{all\_cyc} = \lambda(L_{cyc1} + L_{cyc2}).$$

(13)

where $N$ denotes the number of samples and $\lambda$ is the coefficient used in the computation. The bidirectional cycle consistency loss is computed using the mean squared error between the reconstructed features and real features, ensuring consistency between these features and closing the modality gap.

## 4 Experiments

### 4.1 Datasets

**ThingsEEG dataset [39].** It comprises EEG data from 10 participants collected using the rapid serial visual presentation (RSVP) paradigm. Except for a single reference electrode, the data of 63 electrode channels can be obtained simultaneously, with a sampling frequency set at 1000Hz. Each participant took part in a total of 82,160 trials. The training set, which consists of 1,654 categories, each with 10 images, is repeated 4 times; The test set, on the other hand, comprises 200 categories, each with only 1 image, and is repeated 80 times. In the preprocessing stage, we average all EEG repetitions for each image to reduce noise impact and enhance the quality of the EEG signals. We select EEG time series from 0 ms to 1000 ms post-image onset. The EEG signals are down-sampled to 250 Hz, a frequency that captures the brain electrical activity

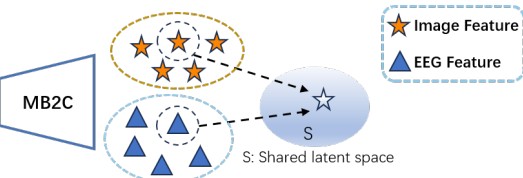

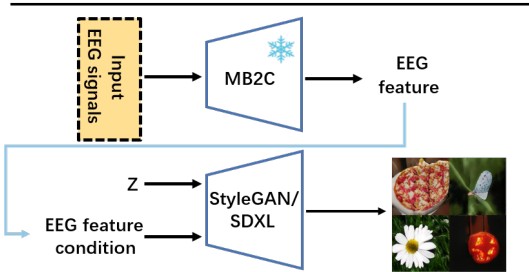

**Figure 2: The decoupled training flow of our image reconstruction using EEG signals. In stage 1, we utilize our proposed method, MB2C, to align EEG-image pairs on the EEGCVPR40 dataset in the shared latent space, as described in Section 3. In stage 2, we utilize EEG information as conditional input to train StyleGAN or finetune SDXL for image reconstruction.**

induced by visual stimuli while significantly reducing the data volume for subsequent analysis.

**EEGCVPR40 dataset [25].** It is a subset of the ImageNet dataset [40], comprising EEG-image pairs from 40 classes. During the experiment, 6 subjects viewed 2000 different images, where each image appeared once for 0.5 seconds, resulting in 12,000 visual-evoked EEG sequences. The EEG data are 128 channels with a sampling rate of 1000 Hz. For EEG data preprocessing, we select EEG data ranging from 20 to 460 ms and discard some faulty samples, resulting in 11,940 EEG-image pairs. To ensure a fair comparison, we have concatenated the data of all participants together, and the splitting of training and testing sets followed the same approach as [7].

### 4.2 Implementation Details

This section describes the implementation details of the visual feature generation network $V = \{G_v, D_v\}$ and the EEG feature generation network $E = \{G_e, D_e\}$. $G_v(\cdot)$ and $G_e(\cdot)$ are implemented with a hidden layer containing 900 hidden units, activated by LeakyReLU, and the output layer has dimensions identical to those of the image encoder. $D_v(\cdot)$ and $D_e(\cdot)$ have two hidden layers with 400 hidden units, applying a dropout rate of 0.25 and LeakyReLU activation. Furthermore, discriminators contain two fully connected layers, where one fully connected layer identifies whether the input features are real or not, and the other fully connected layer classifies the input features into the corresponding categories. In all datasets, the noise vector $z$ is set to a dimension of 100. We select Adam as the optimizer for the brain encoder, brain projector, and image

**Table 1: Classification accuracy (%) of $N$-way Top-$K$ on ThingsEEG dataset. $N$-way Top-$K$ decoding means predicting the Top-$K$ predicted classes out of $N$ novel classes**

| Type | Method | Subject 1 | | Subject 2 | | Subject 3 | | Subject 4 | | Subject 5 | | Subject 6 | | Subject 7 | | Subject 8 | | Subject 9 | | Subject 10 | | Average | |
|---|---|---|---|---|---|---|---|---|---|---|---|---|---|---|---|---|---|---|---|---|---|---|---|
| | | top-1 | top-5 | top-1 | top-5 | top-1 | top-5 | top-1 | top-5 | top-1 | top-5 | top-1 | top-5 | top-1 | top-5 | top-1 | top-5 | top-1 | top-5 | top-1 | top-5 | top-1 | top-5 |
| | | | | | | | | | | | | | | | | | | | | | | | |
| | | Subject dependent: using one subject's training data for training and test data for testing | | | | | | | | | | | | | | | | | | | | | |
| 200-way | BraVL [5] | 6.11 | 17.89 | 4.9 | 14.87 | 5.58 | 17.38 | 4.96 | 15.11 | 4.01 | 13.39 | 6.01 | 18.18 | 6.51 | 20.35 | 8.79 | 23.68 | 4.34 | 13.98 | 7.04 | 19.71 | 5.82 | 17.45 |
| | NICE-GA [6] | 15.2 | 40.1 | 13.9 | 40.1 | 14.7 | 42.7 | 17.6 | 48.9 | 9.0 | 29.7 | 16.4 | 44.4 | 14.9 | 43.1 | 20.3 | 52.1 | 19.6 | 46.7 | | | 15.6 | 42.8 |
| | **MB2C(mixup)** | **23.67** | **56.33** | **22.67** | **50.50** | **26.33** | **60.17** | **34.83** | **67.00** | **21.33** | **53.00** | **31.00** | **62.33** | **25.00** | **54.83** | **39.00** | **69.33** | **27.50** | **59.33** | **33.17** | **70.83** | **28.45** | **60.37** |
| 50-way | BraVL [5] | 14.8 | 41.5 | 12.88 | 39.15 | 15.0 | 40.85 | 12.35 | 36.45 | 10.45 | 33.77 | 15.1 | 41.17 | 15.12 | 42.38 | 20.32 | 49.83 | 10.55 | 34.1 | 16.75 | 43.6 | 14.33 | 40.28 |
| | **MB2C(mixup)** | **41.33** | **83.33** | **38.67** | **82.67** | **48.67** | **84.67** | **56.00** | **84.67** | **39.33** | **70.00** | **54.67** | **86.67** | **45.33** | **80.67** | **68.67** | **89.33** | **53.33** | **89.33** | **58.67** | **90.67** | **50.47** | **84.20** |
| | | Subject independent: using one subject's data for testing and the rest of the subjects' data for training | | | | | | | | | | | | | | | | | | | | | |
| 200-way | BraVL [5] | 2.3 | 7.99 | 1.49 | 6.32 | 1.39 | 5.88 | 1.73 | 6.65 | 1.54 | 5.64 | 1.76 | 7.24 | 2.14 | 8.06 | 2.19 | 7.57 | 1.55 | 6.38 | 2.3 | 8.52 | 1.84 | 7.02 |
| | NICE-SA [6] | 7.0 | 22.6 | 6.6 | 23.2 | 7.5 | 23.7 | 5.4 | 21.4 | 6.4 | 22.2 | 7.5 | 22.5 | 3.8 | 19.1 | 8.5 | 24.4 | 7.4 | 22.3 | 9.8 | 29.6 | 7.0 | 23.1 |
| | **MB2C (mixup)** | **10.50** | **28.17** | **11.33** | **32.83** | **8.83** | **27.67** | **13.67** | **33.50** | **10.67** | **27.50** | **12.17** | **33.17** | **11.50** | **31.83** | **12.00** | **32.17** | **12.17** | **31.33** | **16.17** | **42.17** | **11.90** | **32.03** |
| 50-way | BraVL [5] | 6.38 | 22.98 | 4.98 | 20.7 | 3.92 | 17.8 | 5.6 | 18.6 | 4.67 | 19.38 | 5.65 | 23.08 | 6.25 | 24.12 | 6.02 | 23.9 | 4.58 | 18.7 | 5.85 | 22.8 | 5.39 | 21.2 |
| | **MB2C (mixup)** | **25.33** | **68.00** | **34.67** | **74.00** | **18.00** | **59.33** | **29.33** | **63.33** | **22.00** | **59.33** | **20.00** | **54.67** | **22.67** | **59.33** | **26.67** | **48.67** | **23.33** | **62.67** | **31.33** | **77.33** | **25.33** | **62.67** |

projector with the learning rate set to $2 \times 10^{-4}$. RMSProp is used as the optimizer for the feature generation network with a learning rate of $5 \times 10^{-5}$. Default hyperparameters are set to $\beta = 1.0$, $\alpha = 1.0$, and $\lambda = 500$. In the ThingsEEG dataset, the batch size is 256, and training epochs are set to 200. To prevent model overfitting or underfitting, we adopt an early stopping strategy. In the EEGCVPR40 dataset, we set the batch size to 64, and the training process runs for 2048 epochs. To ensure more accurate experimental results, we conduct each experiment three times with different random seeds and report the average results. All experiments are conducted using PyTorch on a GeForce 3090 GPU. The code has been released at: https://github.com/leeh2213/MB2C

**Table 2: EEG classification accuracy (%) of Top-$K$ on EEGCVPR40 dataset**

| EEG data | Method | top-1 | top-5 | top-10 |
|---|---|---|---|---|
| raw | EEGClip [7] | 79.0 | 96.0 | 98.0 |
| | **MB2C** | **88.73** | **98.24** | **99.14** |
| 5-95 Hz filter | Palazzo et al. [41] | 60.4 | - | - |
| | EEGClip [7] | 64.0 | 86.0 | 92.0 |
| | **MB2C** | **93.74** | **99.18** | **99.63** |

**Table 3: MB2C with different generative models Image reconstruction comparison with baselines in EEGCVPR40 and ThingsEEG datasets.**

| Methods | Datasets | IS ↑ | FID ↓ | KID ↓ | SSIM ↑ | PCC ↑ |
|---|---|---|---|---|---|---|
| Improved-SNGAN [42] | EEGCVPR40 | 5.53 | - | - | - | - |
| DCLS-GAN [43] | EEGCVPR40 | 6.64 | - | - | - | - |
| EEGStyleGAN-ADA [7] | EEGCVPR40 | 10.82 | 174.13 | 0.065 | - | - |
| NEUROIMAGEN [44] | EEGCVPR40 | **33.50** | - | - | 0.249 | - |
| **MB2C-stylegan** | EEGCVPR40 | 12.85 | **153.37** | 0.042 | **0.442** | 0.075 |
| **MB2C-SDXL** | EEGCVPR40 | 21.87 | 171.36 | 0.044 | 0.382 | **0.131** |
| **MB2C-SDXL** | ThingsEEG | 10.19 | 163.94 | 0.027 | 0.333 | 0.188 |

## 4.3 Experimental Results

**$N$-way Zero-shot classification.** We perform $N$-way zero-shot classification tasks on the ThingsEEG dataset following the procedure outlined in Figure 1 for intra-subject and inter-subject experiments. As shown in Table 1, the results indicate that our proposed method possesses excellent neural decoding capabilities. Specifically, in intra-subject experiments, MB2C achieves a top-1 accuracy of 50.47% and an average top-5 accuracy of 84.20% in a 50-way classification task, significantly surpassing chance levels of 2% and 10%, respectively. Even when tested with 200 unseen classes, the model maintains a top-1 accuracy of 28.45% and a top-5 accuracy of 60.37%. In the inter-subject experiments, due to individual differences among subjects, the neural decoding performance falls short of the intra-subject predictions but still significantly outperforms chance levels. We also compare MB2C with state-of-the-art methods, namely BraVL [5] and NICE [6]. Considering NICE with self-attention (NICE-SA) or graph attention (NICE-GA) as two variants that outperform the original NICE model, we select the best-performing model for comparison. The results demonstrate that, regardless of whether the experiments were intra-subject or inter-subject, MB2C outperforms the current state-of-the-art models in both 50-way and 200-way zero-shot classifications.

**Joint representation learning on EEGCVPR40 dataset.** To verify the robustness of MB2C, we conduct experiments on the EEGCVPR40 raw dataset and the EEGCVPR40 filter dataset (5-95Hz). During the model training process, we observe that directly using EEG-visual correlations for classification performed poorly. In the EEGCVPR40 dataset, each EEG-image pair corresponds to having real class labels. To improve the classification of EEG data, we train the EEG Encoder and two projection layers, similar to [7], and then fine-tune the network using EEG data. This refinement results in excellent EEG classification results. As shown in Table 2, our method outperforms the current state-of-the-art models in top-$K$ accuracy, where $K \in \{1, 5, 10\}$, on both the EEGCVPR40 raw dataset and the EEGCVPR40 filter dataset (5-95Hz).

**Image reconstruction.** Here we evaluate the effectiveness of MB2C in the image reconstruction task. Quantitative and qualitative experiments are conducted following the training flow outlined in Figure 2. Due to the small size of the EEGCVPR40 dataset, we train

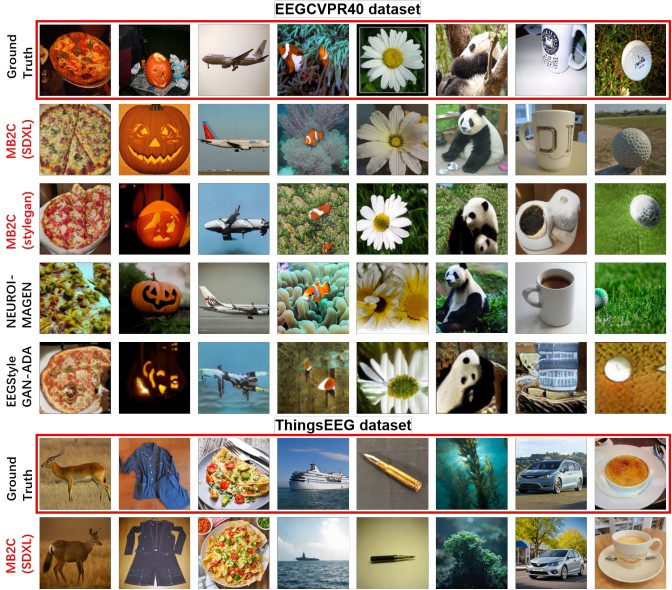

**Figure 3: Image reconstruction conditioned on the EEG for the EEGCVPR40 filtered (5-95 Hz) and ThingsEEG dataset. The rows labeled 'Ground Truth' (highlighted with a red box) represent the ground truth image stimulus. Images with red text correspond to reconstructions using EEG features extracted by the MB2C model, while others correspond to the results presented in the paper.**

StyleGAN from scratch to achieve stability and high fidelity. Additionally, we employ the more powerful generative model SDXL[45], using EEG information as a conditional input for image reconstruction on the EEGCVPR40 filtered (5-95 Hz) and ThingsEEG datasets. Notably, to our knowledge, this is the first image reconstruction on the ThingsEEG dataset.The quantitative comparison results of image reconstruction are listed in Table 3. We utilize five evaluation metrics that shed light on the quality and diversity of the synthesized images (resized to 128×128) to compare our model with some state-of-the-art methods. Inception score (IS), Frechet Inception Distance (FID), Kernel Inception Distance (KID), Per-pixel Correlation Coefficient (PCC), and Structural Similarity Index Measure (SSIM) are calculated based on reconstructed images. As summarized in Table 3, our method outperforms other SOTA baselines. Meanwhile, the results of EEG-conditioned image reconstruction are shown in Figure 3. As can be seen, the quality of the sampling of reconstructed images is semantically similar to the ground truth images, and our method maintains generation consistency in terms of shape, color, and semantics. Additionally, it can be observed that compared to recent work, our approach produces reconstructed results that are visually more similar to the ground truth both in terms of semantics and structure.

## 4.4 Ablation Study

To demonstrate the effectiveness of each component proposed in our method, we introduce two variants of MB2C for comparison:

- InfoNCE-only: It retains only the InfoNCE loss, removing all components related to BCWGAN from the model.
- InfoNCE-CYC-only: The model removes the classification (CLA) loss in MB2C, retaining both the InfoNCE loss and the cycle consistency (CYC) loss.
- no Mixup: The model does not utilize Mixup-based data augmentation.

For a fair comparison, ablation experiments for InfoNCE-only and InfoNCE-CYC-only are conducted with a Mixup ratio coefficient of 0.75 (which yielded the best performance). We will discuss in detail the impact of the Mixup mixing coefficient on model performance in Section 4.6. Additionally, as ZSL is our primary classification task, all ablation experiments are performed on the ThingsEEG dataset. Table 4 displays the performance of each variant. Firstly, it is evident that adding the CYC loss individually on top of contrastive learning can improve overall performance, while jointly training the model with both CYC loss and CLA loss can significantly enhance performance, indicating that the constraint of cycle consistency effectively assists in multimodal alignment. Furthermore, the performance of each variant surpasses that of the methods presented in Table 1, BraVL and NICE-GA, demonstrating the effectiveness of the MB2C approach.

**Table 4: Effects of different components on zero-shot classification accuracy (%) on ThingsEEG datasets**

| Method | 200-way | | 50-way | |
| --- | --- | --- | --- | --- |
| | top-1 | top-5 | top-1 | top-5 |
| CLIP-only | 26.17 | 59.30 | 46.87 | **84.87** |
| CLIP-CYC-only | 27.55 | 59.95 | 50.33 | 84.53 |
| no Mixup | 26.60 | 58.57 | 48.40 | 84.86 |
| **MB2C** | **28.45** | **60.37** | **50.47** | 84.20 |

## 4.5 Smilarity Measures and Visualization Analysis

To visually demonstrate the effectiveness of MB2C, we perform similarity measures and visualization analysis on the ThingsEEG dataset. Firstly, to validate whether the EEG features we extract contain semantic information capable of distinguishing between different image classes, we categorize the image stimuli in the test set into six superclasses: animals, clothes, food, household, tools, and transportation. For instance, "tie" belongs to clothes, and "fork" belongs to tools. Then, we compute the cosine similarity of feature pairs for all subjects in the test set, as shown in Figure 4 (a). The results in Figure 4 (a) indicate that the similarity within the superclass is high, indicating that EEG features effectively reflect the semantic categories of corresponding images.

In addition, we depict the t-SNE [46] plots of visual features and EEG features learned from the training and test sets in Figure 4 (b) and Figure 4 (c). As per our observations, the features of test images exhibit similar distributions to those of training images.

Yayun Wei, Lei Cao, Hao Li, and Yilin Dong

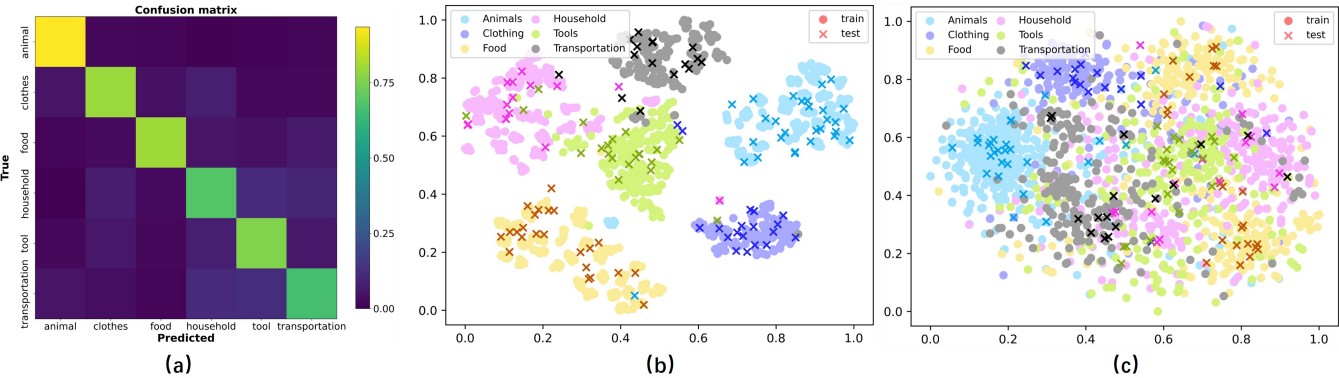

**Figure 4: Cosine similarity and t-SNE visualization on the ThingsEEG dataset. (a) Confusion matrix of feature pairs for all subjects in the test set. All the concepts were reclassified into five categories: animals, clothes, food, household, tools, and transportation. (b) t-SNE visualization of the learned visual feature for the six categories. (c) t-SNE visualization of the learned EEG feature for the six categories.**

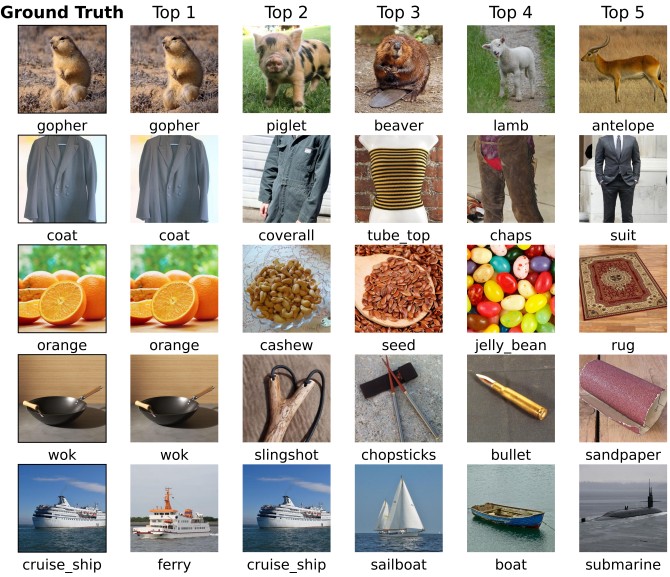

**Figure 5: The ground truth image and the top-5 predicted results for the given EEG signal from the test set of ThingsEEG.**

Moreover, except for a few categories, the spatial distributions of the EEG features in the test and training datasets also maintain high consistency.

In Figure 5, we randomly present the top-5 predicted results from subject 8. We can observe that they are semantically similar to the ground truth; for example, "gopher", "piglet", "beaver", "lamb", and "antelope" all belong to the category of animals.

## 4.6 Sensitivity of Hyper-parameters

We conduct all experiments on the ThingsEEG dataset under the setting of $N$-way Top-$K$ classification task and present the optimal ratio of Mixup samples in Figure 6. Given a fixed batch size $B$ of

256, within a range of ratio in $\{0, 0.25, 0.5, 0.75, 1.00\}$, indicating the choice of 0.75 sampling ratio achieves the best while considering both model performance and training computation (The higher the ratio, the more computations). Thus, we set the ratio to 0.75 under all experiment settings of the ThingsEEG dataset.

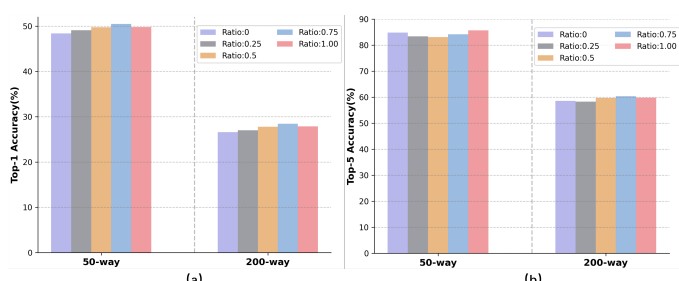

**Figure 6: The ratio of Mixups.**

## 5 Conclusion

In this study, our primary objective is to learn robust visual neural representations from EEG-based brain activity. We propose a novel multimodal framework that combines MB2C with contrastive learning to achieve cross-modal alignment. Notably, MB2C utilizes dual-GAN to generate modality-relevant features and inversely transform them back into corresponding semantic latent spaces, thereby narrowing the modality gap and ensuring embeddings of different modalities with similar semantics reside in the same region of the representation space. Our results demonstrate that decoding seen or even unseen visual categories from EEG signals is potential, and image reconstruction with EEG is also feasible. Lastly, although this paper focuses on EEG and images, we show that MB2C can generalize to other paired modalities. We believe our research holds significant value for practical BCIs and research in multimodal learning.

## Acknowledgments

The work was supported by the National Natural Science Young Foundation of China (Grant No. 62102242) and the Shanghai Education Research Program, China (Grant No. C2022152)

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
