# OpenReview forum: "MB2C: Multimodal Bidirectional Cycle Consistency for Learning Robust Visual Neural Representations"
_acmmm.org/ACMMM/2024/Conference — MM2024 Poster_

### Official Review · Reviewer_Acam · 2024-05-21

**Rating:** 4
**Confidence:** 4

**Summary:**

The author proposes a novel Multimodal Bidirectional Cycle Consistency framework for visual neural representations. In their method, they use InfoNCE loss to train encoders to extract EEG and image embeddings first and feed these features to the generator of BCWGAN for cross-modal feature generation. The experiment result demonstrate the effectiveness of their feature.

**Strengths:**

The introduction of the method is very detailed and the experiment is thorough.

**Limitations:**

1. Figures and tables:

(1) Figure 1 is a little bit complicated,  especially drawing the training objectives of the classification and discriminator into a graph feels a bit redundant. I suggest highlighting Multimodal Bidirectional Cycle Consistency Loss and InfoNCE loss. Do not draw other processes in too much detail.

(2) Some tables have a font that is too large, such as table 4.

2. Experiment result:

(1) Ablation study: it is recommended to correspond to the method, that is, with or without mixed-up data augmentation, infoNCE, and Multimodal Bidirectional Cycle Consistency Loss.

(2) Although the zero-shot classification result is good, the reconstruction results lack comparison with SOTA work in the past three years, such as "Seeing through the Brain: Image Reconstruction of Visual Perception
from Human Brain Signals" and "DreamDiffusion: Generating High-Quality Images from Brain EEG Signals". Also, the reconstruction image in Figure 3 also looks unsatisfactory.

**Suitability:**

3

---

### Official Review · Reviewer_Ci5M · 2024-05-22

**Rating:** 4
**Confidence:** 3

**Summary:**

This paper proposed a novel method to reconstruct images from EEG. The main idea is to enforce a cycle consistency loss between EEG embedding and image embeddings. The authors also utilized contrastive learning with mixup for better generalizability. Evaluations performed on two datasets show superior performance compared with previous work.

**Strengths:**

This task is of great interest for the practical use of portable BCI. The paper is well-written, with rich literature and a detailed description of the method. The cycle consistency loss is novel and useful in this task. Solid experiments were performed to support the proposed method.

**Limitations:**

1. Figure 1 is hard to understand. The author may consider splitting this figure into multiple modules or redesigning this figure.

2. Can the author also provide pixel-level metrics such as PCC and SSIM in addition to the classification-based metrics?

3. Can the author provide more visual results other than Figure 3? It's also better to provide a visual comparison with previous works.

4. I am a bit confused about the multimodal claim. It seems only two modalities are involved, e.g., EEG signals and images. I suggest removing the multimodal claim to avoid confusion if these two are the only modalities involved. Otherwise, all EEG-to-image models will be multimodal.

5. About the mixup-based data augmentation. From Eqn(1), the EEG data are linear combined to generate augmented data. Is it a valid operation? I can understand the mixup in the latent space for image features. But will the linear combination of EEG signals change the characteristics of the EEG signals?

**Suitability:**

3

---

### Official Review · Reviewer_7tZC · 2024-05-23

**Rating:** 2
**Confidence:** 3

**Summary:**

The paper proposes an EEG visual decoding method based on adversarial learning and cycle consistency, called MB2C, and discusses its application in downstream tasks such as classification, image reconstruction, and retrieval.

**Strengths:**

1. Applying adversarial learning techniques to EEG visual decoding is a very interesting idea.

2. Experimental results in downstream tasks indicate the effectiveness of the methods proposed in the paper for EEG visual decoding.

3. The paper is well-written up until the experiments.

**Limitations:**

1. The quality of the generated images is not very high. Why not try more powerful and popular generative models such as SD or SDXL?

2. Since the dataset only contains 40 categories of images, with only 50 images each, I believe that training a new StyleGAN might lead to issues of model overfitting. For example, if the generative model can only produce these 40 categories of images, then the quality of the images entirely depends on how well the model fits these limited data. Consequently, the diversity of the images is fixed, making it inappropriate to assess the outcomes using the IS focusing solely on image quality and diversity. Moreover, since the image data used to train the StyleGAN includes a lot of solid color backgrounds (white, black), overfitting might make generated items like watches or cameras appear very similar to the GT images, reducing the disparity between the generative model and real data distributions. This might lead to better FID and KID results.  Perhaps it would be more appropriate to adopt a method similar to that used in the field of fMRI-to-image generation (use a pre-trained model).

3. On the EEGCVPR dataset, the authors chose to reconstruct images, while on ThingsEEG, they opted for similarity measures, which makes the experiments in the paper appear inconsistent and the motivation vague. Why not perform reconstruction experiments on ThingsEEG or use similarity measures on EEGCVPR? Furthermore, the image reconstruction seems to lack qualitative comparisons with baselines. Considering that the metrics are affected by the GAN trained with limited data, I am unable to determine if the image quality produced by this method is sufficiently better than baselines.

4. Minor error:
It seems that the third and fourth images in the second row of Figure 3 are swapped.

In conclusion, in terms of EEG feature decoding, this is a good work. I understand that the main intention of the authors is to demonstrate the decoding capabilities of their method in downstream tasks, but the paper's section on image generation has methodological and experimental issues, which greatly reduces the papers' quality.

**Suitability:**

3

---

### Official Review · Reviewer_PY7H · 2024-05-24

**Rating:** 4
**Confidence:** 4

**Summary:**

This paper proposes a new framework called Multimodal Bidirectional Circular Consistency (MB2C) for learning robust visual neural representations from electroencephalography (EEG) data. This method uses dual GAN to generate modality-related features and reversely transforms them back into the corresponding semantic latent space to narrow the gap between modalities and ensure that different modalities with similar semantics are embedded in the representation space of the same area. Through experiments on ThingsEEG and EEGCVPR40 data sets, the method performs well in zero-shot classification, EEG classification, and image reconstruction tasks, outperforming existing state-of-the-art models.

**Strengths:**

Innovation: A novel multi-modal bidirectional cycle consistency (MB2C) framework is proposed to innovatively solve the problem of cross-modal alignment.

Experimental results: Extensive experiments on multiple benchmark datasets show that the method outperforms existing methods in zero-shot classification, EEG classification, and image reconstruction tasks.

Comprehensiveness: The paper not only shows the overall effect of the method, but also conducts ablation experiments and hyperparameter sensitivity analysis, and explores the contribution of each component and the influence of parameters in detail.

**Limitations:**

Data dependence: Although the data augmentation method is used in the paper, the model still relies on a large amount of high-quality EEG and image paired data. When data is scarce or of low quality, model performance may be significantly affected, limiting its application in environments with limited data resources.

Incomplete ablation experiments: Although ablation experiments were performed, ablation experiments were not performed on the mixup method, resulting in the inability to fully evaluate the specific contribution of this data enhancement to the overall model performance. This incomplete experimental design limits in-depth understanding and validation of the role of various parts of the model.

Code openness: This method contains multiple complex components (such as double GAN and cycle consistency mechanism), and there are many implementation details.  The paper makes no mention of whether the code is public. If the code is not made public, it will affect the ability of other researchers to reproduce the experimental results and conduct further research on this basis, limiting the academic dissemination and practical application of this method.

**Suitability:**

3

---

### Meta-Review · Area_Chair_ENBm · 2024-07-01

**Recommendation:** Accept (Poster)
**Confidence:** 3

**Metareview:**

The paper proposes a method for learning visual neural representations from EEG data.

The reviewers mention (+) the idea of applying adversarial learning to EEG visual decoding being interesting, and (+) the experiments showing the effectiveness of the method.

There are remaining concerns on (-) using GAN for image generation is not state of the art anymore, and (-) the overall quality of the results. They also mention (-) the authors needing to clarify how the data was filtered across different Hz ranges.

After rebuttal, the reviews are somewhat mixed, but positively tending in their ratings (1x weak accept, 2x borderline accept, and 1x borderline reject). Several reviewers increased their ratings such as one from borderline accept to weak accept and one from weak accept to borderline reject. While some reviewers mention open concerns ("not enough responses"), overall this paper could be worthy to be discussed as a poster for the conference.